# Bayesian Optimization with Gradients

Jian Wu [1]    Matthias Poloczek [2]    Andrew Gordon Wilson [1]    Peter I. Frazier [1]
[1] Cornell University, [2] University of Arizona

## Abstract

Bayesian optimization has been successful at global optimization of expensive-to-evaluate multimodal objective functions. However, unlike most optimization methods, Bayesian optimization typically does not use derivative information. In this paper we show how Bayesian optimization can exploit derivative information to find good solutions with fewer objective function evaluations. In particular, we develop a novel Bayesian optimization algorithm, the derivative-enabled knowledge-gradient (*d-KG*), which is one-step Bayes-optimal, asymptotically consistent, and provides greater one-step value of information than in the derivative-free setting. *d-KG* accommodates noisy and incomplete derivative information, comes in both sequential and batch forms, and can optionally reduce the computational cost of inference through automatically selected retention of a single directional derivative. We also compute the *d-KG* acquisition function and its gradient using a novel fast discretization-free technique. We show *d-KG* provides state-of-the-art performance compared to a wide range of optimization procedures with and without gradients, on benchmarks including logistic regression, deep learning, kernel learning, and k-nearest neighbors.

## 1   Introduction

Bayesian optimization [3, 17] is able to find global optima with a remarkably small number of potentially noisy objective function evaluations. Bayesian optimization has thus been particularly successful for automatic hyperparameter tuning of machine learning algorithms [10, 11, 35, 38], where objectives can be extremely expensive to evaluate, noisy, and multimodal.

Bayesian optimization supposes that the objective function (e.g., the predictive performance with respect to some hyperparameters) is drawn from a prior distribution over functions, typically a Gaussian process (GP), maintaining a posterior as we observe the objective in new places. Acquisition functions, such as expected improvement [15, 17, 28], upper confidence bound [37], predictive entropy search [14] or the knowledge gradient [32], determine a balance between exploration and exploitation, to decide where to query the objective next. By choosing points with the largest acquisition function values, one seeks to identify a global optimum using as few objective function evaluations as possible.

Bayesian optimization procedures do not generally leverage derivative information, beyond a few exceptions described in Sect. 2. By contrast, other types of continuous optimization methods [36] use gradient information extensively. The broader use of gradients for optimization suggests that gradients should also be quite useful in Bayesian optimization: (1) Gradients inform us about the objective's *relative* value as a function of location, which is well-aligned with optimization. (2) In $d$-dimensional problems, gradients provide $d$ distinct pieces of information about the objective's relative value in each direction, constituting $d + 1$ values per query together with the objective value itself. This advantage is particularly significant for high-dimensional problems. (3) Derivative information is available in many applications at little additional cost. Recent work [e.g., 23] makes gradient information available for hyperparameter tuning. Moreover, in the optimization of engineering systems modeled by partial differential equations, which pre-dates most hyperparameter tuning applications [8], adjoint

methods provide gradients cheaply [16, 29]. And even when derivative information is not readily available, we can compute approximative derivatives in parallel through finite differences.

In this paper, we explore the "what, when, and why" of Bayesian optimization with derivative information. We also develop a Bayesian optimization algorithm that effectively leverages gradients in hyperparameter tuning to outperform the state of the art. This algorithm accommodates incomplete and noisy gradient observations, can be used in both the sequential and batch settings, and can optionally reduce the computational overhead of inference by selecting the single most valuable directional derivatives to retain. For this purpose, we develop a new acquisition function, called the derivative-enabled knowledge-gradient (*d-KG*). *d-KG* generalizes the previously proposed batch knowledge gradient method of Wu and Frazier [44] to the derivative setting, and replaces its approximate discretization-based method for calculating the knowledge-gradient acquisition function by a novel faster exact discretization-free method. We note that this discretization-free method is also of interest beyond the derivative setting, as it can be used to improve knowledge-gradient methods for other problem settings. We also provide a theoretical analysis of the *d-KG* algorithm, showing (1) it is one-step Bayes-optimal by construction when derivatives are available; (2) that it provides one-step value greater than in the derivative-free setting, under mild conditions; and (3) that its estimator of the global optimum is asymptotically consistent.

In numerical experiments we compare with state-of-the-art batch Bayesian optimization algorithms with and without derivative information, and the gradient-based optimizer BFGS with full gradients.

We assume familiarity with GPs and Bayesian optimization, for which we recommend Rasmussen and Williams [31] and Shahriari et al. [34] as a review. In Section 2 we begin by describing related work. In Sect. 3 we describe our Bayesian optimization algorithm exploiting derivative information. In Sect. 4 we compare the performance of our algorithm with several competing methods on a collection of synthetic and real problems.

The code for this paper is available at `https://github.com/wujian16/Cornell-MOE`.

## 2   Related Work

Osborne et al. [26] proposes fully Bayesian optimization procedures that use derivative observations to improve the conditioning of the GP covariance matrix. Samples taken near previously observed points use only the derivative information to update the covariance matrix. Unlike our current work, derivative information does not affect the acquisition function. We directly compare with Osborne et al. [26] within the KNN benchmark in Sect. 4.2.

Lizotte [22, Sect. 4.2.1 and Sect. 5.2.4] incorporates derivatives into Bayesian optimization, modeling the derivatives of a GP as in Rasmussen and Williams [31, Sect. 9.4]. Lizotte [22] shows that Bayesian optimization with the expected improvement (EI) acquisition function and complete gradient information at each sample can outperform BFGS. Our approach has six key differences: (i) we allow for noisy and incomplete derivative information; (ii) we develop a novel acquisition function that outperforms EI with derivatives; (iii) we enable batch evaluations; (iv) we implement and compare batch Bayesian optimization with derivatives across several acquisition functions, on benchmarks and new applications such as kernel learning, logistic regression, deep learning and k-nearest neighbors, further revealing empirically where gradient information will be most valuable; (v) we provide a theoretical analysis of Bayesian optimization with derivatives; (vi) we develop a scalable implementation.

Very recently, Koistinen et al. [19] uses GPs with derivative observations for minimum energy path calculations of atomic rearrangements and Ahmed et al. [1] studies expected improvement with gradient observations. In Ahmed et al. [1], a randomly selected directional derivative is retained in each iteration for computational reasons, which is similar to our approach of retaining a single directional derivative, though differs in its random selection in contrast with our value-of-information-based selection. Our approach is complementary to these works.

For batch Bayesian optimization, several recent algorithms have been proposed that choose a set of points to evaluate in each iteration [5, 6, 12, 18, 24, 33, 35, 39]. Within this area, our approach to handling batch observations is most closely related to the batch knowledge gradient (*KG*) of Wu and Frazier [44]. We generalize this approach to the derivative setting, and provide a novel exact method for computing the knowledge-gradient acquisition function that avoids the discretization used in Wu

and Frazier [44]. This generalization improves speed and accuracy, and is also applicable to other knowledge gradient methods in continuous search spaces.

Recent advances improving both access to derivatives and computational tractability of GPs make Bayesian optimization with gradients increasingly practical and timely for discussion.

## 3 Knowledge Gradient with Derivatives

Sect. 3.1 reviews a general approach to incorporating derivative information into GPs for Bayesian optimization. Sect. 3.2 introduces a novel acquisition function *d-KG*, based on the knowledge gradient approach, which utilizes derivative information. Sect. 3.3 computes this acquisition function and its gradient efficiently using a novel fast discretization-free approach. Sect. 3.4 shows that this algorithm provides greater value of information than in the derivative-free setting, is one-step Bayes-optimal, and is asymptotically consistent when used over a discretized feasible space.

### 3.1 Derivative Information

Given an expensive-to-evaluate function $f$, we wish to find $\operatorname{argmin}_{x \in \mathbb{A}} f(x)$, where $\mathbb{A} \subset \mathbb{R}^d$ is the domain of optimization. We place a GP prior over $f \colon \mathbb{A} \to \mathbb{R}$, which is specified by its mean function $\mu \colon \mathbb{A} \to \mathbb{R}$ and kernel function $K \colon \mathbb{A} \times \mathbb{A} \to \mathbb{R}$. We first suppose that for each sample of $x$ we observe the function value and all $d$ partial derivatives, possibly with independent normally distributed noise, and then later discuss relaxation to observing only a single directional derivative.

Since the gradient is a linear operator, the gradient of a GP is also a GP (see also Sect. 9.4 in Rasmussen and Williams [31]), and the function and its gradient follow a multi-output GP with mean function $\tilde{\mu}$ and kernel function $\tilde{K}$ defined below:

$$\tilde{\mu}(x) = (\mu(x), \nabla\mu(x))^T, \quad \tilde{K}(x, x') = \begin{pmatrix} K(x, x') & J(x, x') \\ J(x', x)^T & H(x, x') \end{pmatrix} \tag{3.1}$$

where $J(x, x') = \left( \frac{\partial K(x,x')}{\partial x_1'}, \cdots, \frac{\partial K(x,x')}{\partial x_d'} \right)$ and $H(x, x')$ is the $d \times d$ Hessian of $K(x, x')$.

When evaluating at a point $x$, we observe the noise-obscured function value $y(x)$ and gradient $\nabla y(x)$. Jointly, these observations form a $(d + 1)$-dimensional vector with conditional distribution

$$(y(x), \nabla y(x))^T \mid f(x), \nabla f(x) \sim \mathcal{N}\left( (f(x), \nabla f(x))^T, \operatorname{diag}(\sigma^2(x)) \right), \tag{3.2}$$

where $\sigma^2 \colon \mathbb{A} \to \mathbb{R}_{\geq 0}^{d+1}$ gives the variance of the observational noise. If $\sigma^2$ is not known, we may estimate it from data. The posterior distribution is again a GP. We refer to the mean function of this posterior GP after $n$ samples as $\tilde{\mu}^{(n)}(\cdot)$ and its kernel function as $\tilde{K}^{(n)}(\cdot, \cdot)$. Suppose that we have sampled at $n$ points $X := \{x^{(1)}, x^{(2)}, \cdots, x^{(n)}\}$ and observed $(y, \nabla y)^{(1:n)}$, where each observation consists of the function value and the gradient at $x^{(i)}$. Then $\tilde{\mu}^{(n)}(\cdot)$ and $\tilde{K}^{(n)}(\cdot, \cdot)$ are given by

$$\tilde{\mu}^{(n)}(x) = \tilde{\mu}(x) + \tilde{K}(x, X)$$
$$\left( \tilde{K}(X, X) + \operatorname{diag}\{\sigma^2(x^{(1)}), \cdots, \sigma^2(x^{(n)})\} \right)^{-1} \left( (y, \nabla y)^{(1:n)} - \tilde{\mu}(X) \right)$$
$$\tilde{K}^{(n)}(x, x') = \tilde{K}(x, x') - \tilde{K}(x, X) \left( \tilde{K}(X, X) + \operatorname{diag}\{\sigma^2(x^{(1)}), \cdots, \sigma^2(x^{(n)})\} \right)^{-1} \tilde{K}(X, x'). \tag{3.3}$$

If our observations are incomplete, then we remove the rows and columns in $(y, \nabla y)^{(1:n)}$, $\tilde{\mu}(X)$, $\tilde{K}(\cdot, X)$, $\tilde{K}(X, X)$ and $\tilde{K}(X, \cdot)$ of Eq. (3.3) corresponding to partial derivatives (or function values) that were not observed. If we can observe directional derivatives, then we add rows and columns corresponding to these observations, where entries in $\tilde{\mu}(X)$ and $\tilde{K}(\cdot, \cdot)$ are obtained by noting that a directional derivative is a linear transformation of the gradient.

### 3.2 The *d-KG* Acquisition Function

We propose a novel Bayesian optimization algorithm to exploit available derivative information, based on the knowledge gradient approach [9]. We call this algorithm the *derivative-enabled knowledge gradient* (*d-KG*).

The algorithm proceeds iteratively, selecting in each iteration a batch of $q$ points in $\mathbb{A}$ that has a maximum *value of information* (VOI). Suppose we have observed $n$ points, and recall from Section 3.1 that $\tilde{\mu}^{(n)}(x)$ is the $(d+1)$-dimensional vector giving the posterior mean for $f(x)$ and its $d$ partial derivatives at $x$. Sect. 3.1 discusses how to remove the assumption that all $d+1$ values are provided.

The expected value of $f(x)$ under the posterior distribution is $\tilde{\mu}_1^{(n)}(x)$. If after $n$ samples we were to make an irrevocable (risk-neutral) decision now about the solution to our overarching optimization problem and receive a loss equal to the value of $f$ at the chosen point, we would choose $\operatorname{argmin}_{x\in\mathbb{A}}\tilde{\mu}_1^{(n)}(x)$ and suffer conditional expected loss $\min_{x\in\mathbb{A}}\tilde{\mu}_1^{(n)}(x)$. Similarly, if we made this decision after $n+q$ samples our conditional expected loss would be $\min_{x\in\mathbb{A}}\tilde{\mu}_1^{(n+q)}(x)$. Therefore, we define the *d-KG* factor for a given set of $q$ candidate points $z^{(1:q)}$ as

$$d\text{-}KG(z^{(1:q)}) \quad = \quad \min_{x\in\mathbb{A}}\tilde{\mu}_1^{(n)}(x) - \mathbb{E}_n\left[\min_{x\in\mathbb{A}}\tilde{\mu}_1^{(n+q)}(x)\ \middle|\ x^{((n+1):(n+q))} = z^{(1:q)}\right],$$
(3.4)

where $\mathbb{E}_n\left[\cdot\right]$ is the expectation taken with respect to the posterior distribution after $n$ evaluations, and the distribution of $\tilde{\mu}_1^{(n+q)}(\cdot)$ under this posterior marginalizes over the observations $\left(y(z^{(1:q)}), \nabla y(z^{(1:q)})\right) = \left(y(z^{(i)}), \nabla y(z^{(i)}) : i = 1, \ldots, q\right)$ upon which it depends. We subsequently refer to Eq. (3.4) as the *inner optimization problem*.

The *d-KG* algorithm then seeks to evaluate the batch of points next that maximizes the *d-KG* factor,

$$\max_{z^{(1:q)}\subset\mathbb{A}} d\text{-}KG(z^{(1:q)}).$$
(3.5)

We refer to Eq. (3.5) as the *outer optimization problem*. *d-KG* solves the outer optimization problem using the method described in Section 3.3.

The *d-KG* acquisition function differs from the batch knowledge gradient acquisition function in Wu and Frazier [44] because here the posterior mean $\tilde{\mu}_1^{(n+q)}(x)$ at time $n+q$ depends on $\nabla y(z^{(1:q)})$. This in turn requires calculating the distribution of these gradient observations under the time-$n$ posterior and marginalizing over them. Thus, the *d-KG* algorithm differs from *KG* not just in that gradient observations change the posterior, but also in that the prospect of *future* gradient observations changes the acquisition function. An additional major distinction from Wu and Frazier [44] is that *d-KG* employs a novel discretization-free method for computing the acquisition function (see Section 3.3).

Fig. 1 illustrates the behavior of *d-KG* and *d-EI* on a 1-d example. *d-EI* generalizes expected improvement (EI) to batch acquisition with derivative information [22]. *d-KG* clearly chooses a better point to evaluate than *d-EI*.

Including all $d$ partial derivatives can be computationally prohibitive since GP inference scales as $\mathcal{O}(n^3(d+1)^3)$. To overcome this challenge while retaining the value of derivative observations, we can include only one directional derivative from each iteration in our inference. *d-KG* can naturally decide which derivative to include, and can adjust our choice of where to best sample given that we observe more limited information. We define the *d-KG* acquisition function for observing only the function value and the derivative with direction $\theta$ at $z^{(1:q)}$ as

$$d\text{-}KG(z^{(1:q)}, \theta) = \min_{x\in\mathbb{A}}\tilde{\mu}_1^{(n)}(x) - \mathbb{E}_n\left[\min_{x\in\mathbb{A}}\tilde{\mu}_1^{(n+q)}(x)\ \middle|\ x^{((n+1):(n+q))} = z^{(1:q)}; \theta\right].$$
(3.6)

where conditioning on $\theta$ is here understood to mean that $\tilde{\mu}_1^{(n+q)}(x)$ is the conditional mean of $f(x)$ given $y(z^{(1:q)})$ and $\theta^T\nabla y(z^{(1:q)}) = (\theta^T\nabla y(z^{(i)}) : i = 1, \ldots, q)$. The full algorithm is as follows.

---
**Algorithm 1** *d-KG* with Relevant Directional Derivative Detection
---
1: **for** $t = 1$ to $N$ **do**
2:     $(z^{(1:q)^*}, \theta^*) = \operatorname{argmax}_{z^{(1:q)}, \theta} d\text{-}KG(z^{(1:q)}, \theta)$
3:     Augment data with $y(z^{(1:q)^*})$ and $\theta^{*T}\nabla y(z^{(1:q)^*})$. Update our posterior on $(f(x), \nabla f(x))$.
4: **end for**
    Return $x^* = \operatorname{argmin}_{x\in\mathbb{A}}\tilde{\mu}_1^{Nq}(x)$
---

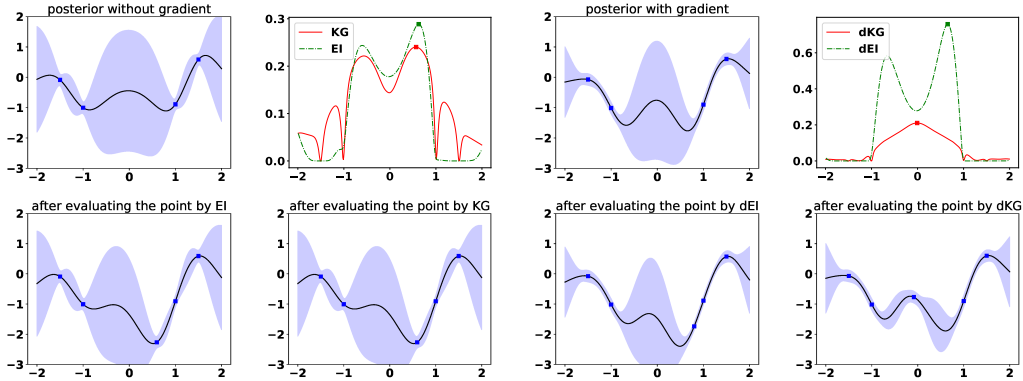

Figure 1: KG [44] and EI [39] refer to acquisition functions without gradients. *d-KG* and *d-EI* refer to the counterparts with gradients. The topmost plots show (1) the posterior surfaces of a function sampled from a one dimensional GP without and with incorporating observations of the gradients. The posterior variance is smaller if the gradients are incorporated; (2) the utility of sampling each point under the value of information criteria of KG (*d-KG*) and EI (*d-EI*) in both settings. If no derivatives are observed, both KG and EI will query a point with high potential gain (i.e. a small expected function value). On the other hand, when gradients are observed, *d-KG* makes a considerably better sampling decision, whereas *d-EI* samples essentially the same location as *EI*. The plots in the bottom row depict the posterior surface *after* the respective sample. Interestingly, KG benefits more from observing the gradients than EI (the last two plots): *d-KG*'s observation yields accurate knowledge of the optimum's location, while *d-EI*'s observation leaves substantial uncertainty.

### 3.3 Efficient Exact Computation of *d-KG*

Calculating and maximizing *d-KG* is difficult when $\mathbb{A}$ is continuous because the term $\min_{x \in \mathbb{A}} \tilde{\mu}_1^{(n+q)}(x)$ in Eq. (3.6) requires optimizing over a continuous domain, and then we must integrate this optimal value through its dependence on $y(z^{(1:q)})$ and $\theta^T \nabla y(z^{(1:q)})$. Previous work on the knowledge gradient in continuous domains [30, 32, 44] approaches this computation by taking minima within expectations not over the full domain $\mathbb{A}$ but over a discretized finite approximation. This approach supports analytic integration in Scott et al. [32] and Poloczek et al. [30], and a sampling-based scheme in Wu and Frazier [44]. However, the discretization in this approach introduces error and scales poorly with the dimension of $\mathbb{A}$.

Here we propose a novel method for calculating an unbiased estimator of the gradient of *d-KG* which we then use within stochastic gradient ascent to maximize *d-KG*. This method avoids discretization, and thus is exact. It also improves speed significantly over a discretization-based scheme.

In Section A of the supplement we show that the *d-KG* factor can be expressed as

$$d\text{-}KG(z^{(1:q)}, \theta) = \mathbb{E}_n \left[ \min_{x \in \mathbb{A}} \hat{\mu}_1^{(n)}(x) - \min_{x \in \mathbb{A}} \left( \hat{\mu}_1^{(n)}(x) + \hat{\sigma}_1^{(n)}(x, \theta, z^{(1:q)})W \right) \right], \qquad (3.7)$$

where $\hat{\mu}^{(n)}$ is the mean function of $(f(x), \theta^T \nabla f(x))$ after $n$ evaluations, $W$ is a $2q$ dimensional standard normal random column vector and $\hat{\sigma}_1^{(n)}(x, \theta, z^{(1:q)})$ is the first row of a $2 \times 2q$ dimensional matrix, which is related to the kernel function of $(f(x), \theta^T \nabla f(x))$ after $n$ evaluations with an exact form specified in (A.2) of the supplement.

Under sufficient regularity conditions [21], one can interchange the gradient and expectation operators,

$$\nabla d\text{-}KG(z^{(1:q)}, \theta) = -\mathbb{E}_n \left[ \nabla \min_{x \in \mathbb{A}} \left( \hat{\mu}_1^{(n)}(x) + \hat{\sigma}_1^{(n)}(x, \theta, z^{(1:q)})W \right) \right],$$

where here the gradient is with respect to $z^{(1:q)}$ and $\theta$. If $(x, z^{(1:q)}, \theta) \mapsto \left( \hat{\mu}_1^{(n)}(x) + \hat{\sigma}_1^{(n)}(x, \theta, z^{(1:q)})W \right)$ is continuously differentiable and $\mathbb{A}$ is compact, the envelope theorem [25] implies

$$\nabla d\text{-}KG(z^{(1:q)}, \theta) = -\mathbb{E}_n \left[ \nabla \left( \hat{\mu}_1^{(n)}(x^*(W)) + \hat{\sigma}_1^{(n)}(x^*(W), \theta, z^{(1:q)})W \right) \right], \qquad (3.8)$$

where $x^*(W) \in \arg\min_{x \in \mathbb{A}} \left( \hat{\mu}_1^{(n)}(x) + \hat{\sigma}_1^{(n)}(x, \theta, z^{(1:q)})W \right)$. To find $x^*(W)$, one can utilize a multi-start gradient descent method since the gradient is analytically available for the objective

$\hat{\mu}_1^{(n)}(x) + \hat{\sigma}_1^{(n)}(x, \theta, z^{(1:q)})W$. Practically, we find that the learning rate of $l_t^{\text{inner}} = 0.03/t^{0.7}$ is robust for finding $x^*(W)$.

The expression (3.8) implies that $\nabla\left(\hat{\mu}_1^{(n)}(x^*(W)) + \hat{\sigma}_1^{(n)}(x^*(W), \theta, z^{(1:q)})W\right)$ is an unbiased estimator of $\nabla d\text{-}KG(z^{(1:q)}, \theta, \mathbb{A})$, when the regularity conditions it assumes hold. We can use this unbiased gradient estimator within stochastic gradient ascent [13], optionally with multiple starts, to solve the outer optimization problem $\text{argmax}_{z^{(1:q)},\theta} d\text{-}KG(z^{(1:q)}, \theta)$ and can use a similar approach when observing full gradients to solve (3.5). For the outer optimization problem, we find that the learning rate of $l_t^{\text{outer}} = 10 l_t^{\text{inner}}$ performs well over all the benchmarks we tested.

**Bayesian Treatment of Hyperparameters.** We adopt a fully Bayesian treatment of hyperparameters similar to Snoek et al. [35]. We draw $M$ samples of hyperparameters $\phi^{(i)}$ for $1 \leq i \leq M$ via the `emcee` package [7] and average our acquisition function across them to obtain

$$d\text{-}KG_{\text{Integrated}}(z^{(1:q)}, \theta) = \frac{1}{M} \sum_{i=1}^{M} d\text{-}KG(z^{(1:q)}, \theta; \phi^{(i)}), \qquad (3.9)$$

where the additional argument $\phi^{(i)}$ in $d\text{-}KG$ indicates that the computation is performed conditioning on hyperparameters $\phi^{(i)}$. In our experiments, we found this method to be computationally efficient and robust, although a more principled treatment of unknown hyperparameters within the knowledge gradient framework would instead marginalize over them when computing $\tilde{\mu}^{(n+q)}(x)$ and $\tilde{\mu}^{(n)}$.

### 3.4 Theoretical Analysis

Here we present three theoretical results giving insight into the properties of *d-KG*, with proofs in the supplementary material. For the sake of simplicity, we suppose all partial derivatives are provided to *d-KG*. Similar results hold for *d-KG* with relevant directional derivative detection. We begin by stating that the value of information (VOI) obtained by *d-KG* exceeds the VOI that can be achieved in the derivative-free setting.

**Proposition 1.** *Given identical posteriors $\tilde{\mu}^{(n)}$,*

$$d\text{-}KG(z^{(1:q)}) \geq KG(z^{(1:q)}),$$

*where KG is the batch knowledge gradient acquisition function without gradients proposed by Wu and Frazier [44]. This inequality is strict under mild conditions (see Sect. B in the supplement).*

Next, we show that *d-KG* is one-step Bayes-optimal by construction.

**Proposition 2.** *If only one iteration is left and we can observe both function values and partial derivatives, then d-KG is Bayes-optimal among all feasible policies.*

As a complement to the one-step optimality, we show that *d-KG* is asymptotically consistent if the feasible set $\mathbb{A}$ is *finite*. Asymptotic consistency means that *d-KG* will choose the correct solution when the number of samples goes to infinity.

**Theorem 1.** *If the function $f(x)$ is sampled from a GP with known hyperparameters, the d-KG algorithm is asymptotically consistent, i.e.*

$$\lim_{N \to \infty} f(x^*(d\text{-}KG, N)) = \min_{x \in \mathbb{A}} f(x)$$

*almost surely, where $x^*(d\text{-}KG, N)$ is the point recommended by d-KG after $N$ iterations.*

## 4 Experiments

We evaluate the performance of the proposed algorithm *d-KG* with relevant directional derivative detection (Algorithm 1) on six standard synthetic benchmarks (see Fig. 2). Moreover, we examine its ability to tune the hyperparameters for the weighted k-nearest neighbor metric, logistic regression, deep learning, and for a spectral mixture kernel (see Fig. 3).

We provide an easy-to-use `Python` package with the core written in `C++`, available at `https://github.com/wujian16/Cornell-MOE`.

We compare *d-KG* to several state-of-the-art methods: (1) The batch expected improvement method (*EI*) of Wang et al. [39] that does not utilize derivative information and an extension of *EI* that incorporates derivative information denoted *d-EI*. *d-EI* is similar to Lizotte [22] but handles incomplete gradients and supports batches. (2) The batch GP-UCB-PE method of Contal et al. [5] that does not utilize derivative information, and an extension that does. (3) The batch knowledge gradient algorithm without derivative information (*KG*) of Wu and Frazier [44]. Moreover, we generalize the method of Osborne et al. [26] to batches and evaluate it on the KNN benchmark. All of the above algorithms allow incomplete gradient observations. In benchmarks that provide the full gradient, we additionally compare to the gradient-based method L-BFGS-B provided in `scipy`. We suppose that the objective function $f$ is drawn from a Gaussian process $GP(\mu, \Sigma)$, where $\mu$ is a constant mean function and $\Sigma$ is the squared exponential kernel. We sample $M = 10$ sets of hyperparameters by the `emcee` package [7].

Recall that the immediate regret is defined as the loss with respect to a global optimum. The plots for synthetic benchmark functions, shown in Fig. 2, report the log10 of immediate regret of the solution that each algorithm would pick as a function of the number of function evaluations. Plots for other experiments report the objective value of the solution instead of the immediate regret. Error bars give the mean value plus and minus one standard deviation. The number of replications is stated in each benchmark's description.

## 4.1 Synthetic Test Functions

We evaluate all methods on six test functions chosen from Bingham [2]. To demonstrate the ability to benefit from *noisy derivative information*, we sample additive normally distributed noise with zero mean and standard deviation $\sigma = 0.5$ for both the objective function and its partial derivatives. $\sigma$ is unknown to the algorithms and must be estimated from observations. We also investigate how incomplete gradient observations affect algorithm performance. We also experiment with two different batch sizes: we use a batch size $q = 4$ for the Branin, Rosenbrock, and Ackley functions; otherwise, we use a batch size $q = 8$. Fig. 2 summarizes the experimental results.

**Functions with Full Gradient Information.** For 2d Branin on domain $[-5, 15] \times [0, 15]$, 5d Ackley on $[-2, 2]^5$, and 6d Hartmann function on $[0, 1]^6$, we assume that the full gradient is available.

Looking at the results for the Branin function in Fig. 2, *d-KG* outperforms its competitors after 40 function evaluations and obtains the best solution overall (within the limit of function evaluations). BFGS makes faster progress than the Bayesian optimization methods during the first 20 evaluations, but subsequently stalls and fails to obtain a competitive solution. On the Ackley function *d-EI* makes fast progress during the first 50 evaluations but also fails to make subsequent progress. Conversely, *d-KG* requires about 50 evaluations to improve on the performance of *d-EI*, after which *d-KG* achieves the best overall performance again. For the Hartmann function *d-KG* clearly dominates its competitors over all function evaluations.

**Functions with Incomplete Derivative Information.** For the 3d Rosenbrock function on $[-2, 2]^3$ we only provide a noisy observation of the third partial derivative. Both *EI* and *d-EI* get stuck early. *d-KG* on the other hand finds a near-optimal solution after ∼50 function evaluations; *KG*, without derivatives, catches up after ∼75 evaluations and performs comparably afterwards. The 4d Levy benchmark on $[-10, 10]^4$, where the fourth partial derivative is observable with noise, shows a different ordering of the algorithms: *EI* has the best performance, beating even its formulation that uses derivative information. One explanation could be that the smoothness and regular shape of the function surface benefits this acquisition criteria. For the 8d Cosine mixture function on $[-1, 1]^8$ we provide two noisy partial derivatives. *d-KG* and UCB with derivatives perform better than EI-type criterion, and achieve the best performances, with *d-KG* beating UCB with derivatives slightly.

In general, we see that *d-KG* successfully exploits noisy derivative information and has the best overall performance.

## 4.2 Real-World Test Functions

**Weighted k-Nearest Neighbor.** Suppose a cab company wishes to predict the duration of trips. Clearly, the duration not only depends on the endpoints of the trip, but also on the day and time.

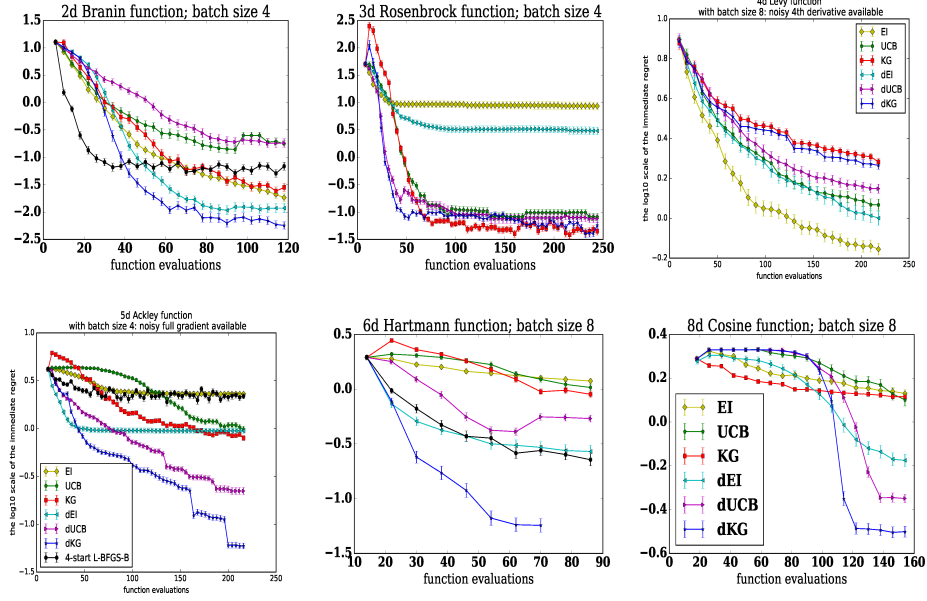

Figure 2: The average performance of 100 replications (the log10 of the immediate regret vs. the number of function evaluations). *d-KG* performs significantly better than its competitors for all benchmarks except Levy funcion. In Branin and Hartmann, we also plot black lines, which is the performance of BFGS.

In this benchmark we tune a weighted k-nearest neighbor (KNN) metric to optimize predictions of these durations, based on historical data. A trip is described by the pick-up time $t$, the pick-up location $(p_1, p_2)$, and the drop-off point $(d_1, d_2)$. Then the estimate of the duration is obtained as a weighted average over all trips $D_{m,t}$ in our database that happened in the time interval $t \pm m$ minutes, where $m$ is a tunable hyperparameter: Prediction$(t, p_1, p_2, d_1, d_2) = (\sum_{i \in D_{m,t}} \text{duration}_i \times \text{weight}(i))/(\sum_{i \in D_{m,t}} \text{weight}(i))$. The weight of trip $i \in D_{m,t}$ in this prediction is given by weight$(i) = ((t - t^i)^2/l_1^2 + (p_1 - p_1^i)^2/l_2^2 + (p_2 - p_2^i)^2/l_3^2 + (d_1 - d_1^i)^2/l_4^2 + (d_2 - d_2^i)^2/l_5^2)^{-1}$,

where $(t^i, p_1^i, p_2^i, d_1^i, d_2^i)$ are the respective parameter values for trip $i$, and $(l_1, l_2, l_3, l_4, l_5)$ are tunable hyperparameters. Thus, we have 6 hyperparameters to tune: $(m, l_1, l_2, l_3, l_4, l_5)$. We choose $m$ in $[30, 200]$, $l_1^2$ in $[10^1, 10^8]$, and $l_2^2, l_3^2, l_4^2, l_5^2$ each in $[10^{-8}, 10^{-1}]$.

We use the yellow cab NYC public data set from June 2016, sampling 10000 records from June 1 – 25 as training data and 1000 trip records from June 26 – 30 as validation data. Our test criterion is the root mean squared error (RMSE), for which we compute the partial derivatives on the validation dataset with respect to the hyperparameters $(l_1, l_2, l_3, l_4, l_5)$, while the hyperparameter $m$ is not differentiable. In Fig. 3 we see that *d-KG* overtakes the alternatives, and that UCB and KG acquisition functions also benefit from exploiting derivative information.

**Kernel Learning.** Spectral mixture kernels [40] can be used for flexible kernel learning to enable long-range extrapolation. These kernels are obtained by modeling a spectral density by a mixture of Gaussians. While any stationary kernel can be described by a spectral mixture kernel with a particular setting of its hyperparameters, initializing and learning these parameters can be difficult. Although we have access to an analytic closed form of the (marginal likelihood) objective, this function is (i) expensive to evaluate and (ii) highly multimodal. Moreover, (iii) derivative information is available. Thus, learning flexible kernel functions is a perfect candidate for our approach.

The task is to train a 2-component spectral mixture kernel on an airline data set [40]. We must determine the mixture weights, means, and variances, for each of the two Gaussians. Fig. 3 summarizes performance for batch size $q = 8$. BFGS is sensitive to its initialization and human intervention and is often trapped in local optima. *d-KG*, on other hand, more consistently finds a good solution, and obtains the best solution of all algorithms (within the step limit). Overall, we observe that gradient information is highly valuable in performing this kernel learning task.

**Logistic Regression and Deep Learning.** We tune logistic regression and a feedforward neural network with 2 hidden layers on the MNIST dataset [20], a standard classification task for handwritten digits. The training set contains 60000 images, the test set 10000. We tune 4 hyperparameters for logistic regression: the $\ell_2$ regularization parameter from 0 to 1, learning rate from 0 to 1, mini batch size from 20 to 2000 and training epochs from 5 to 50. The first derivatives of the first two parameters can be obtained via the technique of Maclaurin et al. [23]. For the neural network, we additionally tune the number of hidden units in $[50, 500]$.

Fig. 3 reports the mean and standard deviation of the mean cross-entropy loss (or its log scale) on the test set for 20 replications. *d-KG* outperforms the other approaches, which suggests that derivative information is helpful. Our algorithm proves its value in tuning a deep neural network, which harmonizes with research computing the gradients of hyperparameters [23, 27].

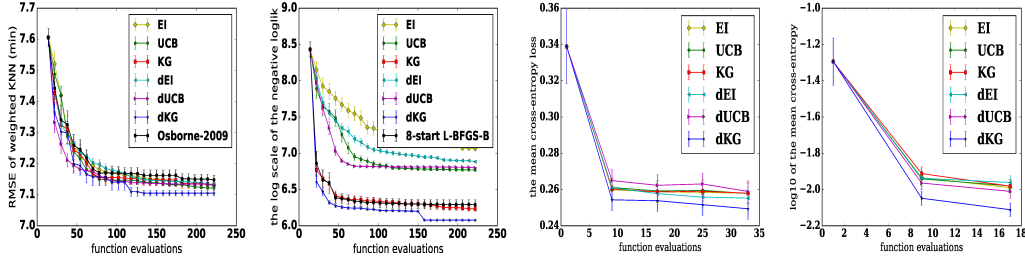

Figure 3: Results for the weighted KNN benchmark, the spectral mixture kernel benchmark, logistic regression and deep neural network (from left to right), all with batch size 8 and averaged over 20 replications.

# 5 Discussion

Bayesian optimization is successfully applied to low dimensional problems where we wish to find a good solution with a very small number of objective function evaluations. We considered several such benchmarks, as well as logistic regression, deep learning, kernel learning, and k-nearest neighbor applications. We have shown that in this context derivative information can be extremely useful: we can greatly decrease the number of objective function evaluations, especially when building upon the knowledge gradient acquisition function, even when derivative information is noisy and only available for some variables.

Bayesian optimization is increasingly being used to automate parameter tuning in machine learning, where objective functions can be extremely expensive to evaluate. For example, the parameters to learn through Bayesian optimization could even be the hyperparameters of a deep neural network. We expect derivative information with Bayesian optimization to help enable such promising applications, moving us towards fully automatic and principled approaches to statistical machine learning.

In the future, one could combine derivative information with flexible deep projections [43], and recent advances in scalable Gaussian processes for $\mathcal{O}(n)$ training and $\mathcal{O}(1)$ test time predictions [41, 42]. These steps would help make Bayesian optimization applicable to a much wider range of problems, wherever standard gradient based optimizers are used – even when we have analytic objective functions that are not expensive to evaluate – while retaining faster convergence and robustness to multimodality.

**Acknowledgments**

Wilson was partially supported by NSF IIS-1563887. Frazier, Poloczek, and Wu were partially supported by NSF CAREER CMMI-1254298, NSF CMMI-1536895, NSF IIS-1247696, AFOSR FA9550-12-1-0200, AFOSR FA9550-15-1-0038, and AFOSR FA9550-16-1-0046.

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
