[Supplementary Material · nips-supplementary.pdf]

# Bayesian Optimization with Gradients

*Supplementary Material*

**Jian Wu**[1]    **Matthias Poloczek**[2]    **Andrew Gordon Wilson**[1]    **Peter I. Frazier**[1]
[1]Cornell University, [2]University of Arizona

## A    The Computation of *d-KG* and its Gradient: Additional Details

In this section, we show additional details in Sect. 3.3 of the main document: how to provide unbiased estimators of $d\text{-}KG(z^{(1:q)}, \theta)$ and its gradient. It is well-known that if $\tilde{\mu}^{(n)}$ and $\tilde{K}^{(n)}$ are the mean and the kernel function respectively of the posterior of $(f(x), \nabla f(x))^T$ after evaluating $n$ points, then $(f(x), \theta^T \nabla f(x)))^T$ follows a bivariate Gaussian process with the mean function $\hat{\mu}^{(n)}$ and the kernel function $\hat{K}^{(n)}$ as follows

$$\hat{\mu}^{(n)}(x) = \begin{pmatrix} 1 & 0_{1 \times d} \\ 0 & \theta^T \end{pmatrix} \tilde{\mu}^{(n)}(x) \text{ and } \hat{K}^{(n)}(x^1, x^2) = \begin{pmatrix} 1 & 0_{1 \times d} \\ 0 & \theta^T \end{pmatrix} \tilde{K}^{(n)}(x^1, x^2) \begin{pmatrix} 1 & 0_{1 \times d} \\ 0 & \theta^T \end{pmatrix}^T.$$

Analogously, the $(y(x), \theta^T \nabla y(x))^T$ is also subject to noise,

$$\left(y(x), \theta^T \nabla y(x)\right)^T \Big| f(x), \theta^T \nabla f(x) \sim \mathcal{N}\left(\left(f(x), \theta^T \nabla f(x)\right)^T, \text{diag}(\hat{\sigma}^2(x))\right),$$

where $\hat{\sigma}^2(x) = \begin{pmatrix} 1 & 0_{1 \times d} \\ 0 & (\theta^T)^2 \end{pmatrix} \sigma^2(x)$. Following Wu and Frazier [44], we express $\hat{\mu}^{(n+q)}(x)$ as

$$\begin{aligned}
\hat{\mu}^{(n+q)}(x) &= \hat{\mu}^{(n)}(x) + \hat{K}^{(n)}(x, z^{(1:q)}) \left(\hat{K}^{(n)}(z^{(1:q)}, z^{(1:q)})\right. \\
&\quad \left. + \text{diag}\{\hat{\sigma}^2(z^{(1)}), \cdots, \hat{\sigma}^2(z^{(q)})\}\right)^{-1} \left((y, \theta^T \nabla y)(z^{(1:q)}) - \hat{\mu}^{(n)}(z^{(1:q)})\right).
\end{aligned}$$

Conditioning on $z^{(1:q)}$ and the knowledge after $n$ evaluations, we have $(y, \theta^T \nabla y)(z^{(1:q)})$ is normally distributed with mean $\hat{\mu}^{(n)}(z^{(1:q)})$ and covariance matrix $\hat{K}^{(n)}(z^{(1:q)}, z^{(1:q)}) + \text{diag}\{\hat{\sigma}^2(z^{(1)}), \cdots, \hat{\sigma}^2(z^{(q)})\}$ where the function $(y, \theta^T \nabla y) : \mathbb{R}^d \to \mathbb{R}^2$ maps the sample to its function and the directional derivative observation at direction $\theta$. Thus, we can rewrite $\hat{\mu}^{(n+q)}(x)$ as

$$\hat{\mu}^{(n+q)}(x) = \hat{\mu}^{(n)}(x) + \hat{\sigma}^{(n)}(x, \theta, z^{(1:q)}) Z_{2q}, \tag{A.1}$$

where $Z_{2q}$ is a $2q$-dimensional standard normal vector and

$$\hat{\sigma}^{(n)}(x, \theta, z^{(1:q)}) = \hat{K}^{(n)}(x, z^{(1:q)}) \left(\hat{D}^{(n)}(z^{(1:q)})^T\right)^{-1}. \tag{A.2}$$

Here $\hat{D}^{(n)}(z^{(1:q)})$ is the Cholesky factor of the covariance matrix $\hat{K}^{(n)}(z^{(1:q)}, z^{(1:q)}) + \text{diag}\{\hat{\sigma}^2(z^{(1)}), \cdots, \hat{\sigma}^2(z^{(q)})\}$. Now we can follow Sect. 3.3 of the main document to estimate $d\text{-}KG(z^{(1:q)}, \theta)$ and its gradient.

## B    Proof of Proposition 1 and Proposition 2

*Proof of Proposition 1.* Recall that we start with the same posterior $\tilde{\mu}^{(n)}$. Then

$$\begin{aligned}
d\text{-}KG(z^{(1:q)}) &= \min_{x \in \mathbb{A}} \tilde{\mu}_1^{(n)}(x) - \mathbb{E}_n\left[\min_{x \in \mathbb{A}} \mathbb{E}_n\left[y(x)|y(z^{(1:q)}), \nabla y(z^{(1:q)})\right] \Big| z^{(1:q)}\right], \\
&= \min_{x \in \mathbb{A}} \tilde{\mu}_1^{(n)}(x) - \mathbb{E}_n\left[\mathbb{E}_n\left[\min_{x \in \mathbb{A}} \mathbb{E}_n\left[y(x)|y(z^{(1:q)}), \nabla y(z^{(1:q)})\right] \big| y(z^{(1:q)})\right] \Big| z^{(1:q)}\right], \\
&\geq \min_{x \in \mathbb{A}} \tilde{\mu}_1^{(n)}(x) - \mathbb{E}_n\left[\min_{x \in \mathbb{A}} \mathbb{E}_n\left[\mathbb{E}_n\left[y(x)|y(z^{(1:q)}), \nabla y(z^{(1:q)})\right] \big| y(z^{(1:q)})\right] \Big| z^{(1:q)}\right], \\
&= \min_{x \in \mathbb{A}} \tilde{\mu}_1^{(n)}(x) - \mathbb{E}_n\left[\min_{x \in \mathbb{A}} \mathbb{E}_n\left[y(x)|y(z^{(1:q)})\right] \Big| z^{(1:q)}\right], \\
&= KG(z^{(1:q)}), \tag{B.1}
\end{aligned}$$

where recall that $y(x)$ is the observed function value at $x$, and $\nabla y(x)$ are the $d$ derivative observations at $x$. The inequality above holds due to Jensen's inequality.

Now we will show that the inequality is strict when $x^*(y(z^{(1:q)}), \nabla y(z^{(1:q)}))$ depends on $\nabla y(z^{(1:q)})$, where $x^*(y(z^{(1:q)}), \nabla y(z^{(1:q)})) \in \arg\min_{x \in \mathcal{A}} \mathbb{E}_n \left[ y(x) | y(z^{(1:q)}), \nabla y(z^{(1:q)}) \right]$. Equality holds only if there exists a set $\mathcal{S}_1$ such that: (1) $P(y(z^{(1:q)}) \in \mathcal{S}_1) = 1$; (2) for any given $y(z^{(1:q)}) \in \mathcal{S}_1$, $\min_{x \in \mathbb{A}} \mathbb{E}_n \left[ y(x) | y(z^{(1:q)}), \nabla y(z^{(1:q)}) \right]$ is a linear function of $\nabla y(z^{(1:q)})$ for all $\nabla y(z^{(1:q)})$ in a set $\mathcal{S}_2$ (allowed to depend on $y(z^{(1:q)})$) such that $P(\nabla y(z^{(1:q)}) \in \mathcal{S}_2 \mid y(z^{(1:q)})) = 1$.

By (3.3), we can express $\tilde{\mu}^{(n+q)}(x)$ as

$$
\mathbb{E}_n \left[ y(x) | y(z^{(1:q)}), \nabla y(z^{(1:q)}) \right]
$$
$$
= \tilde{\mu}^{(n+q)}(x)
$$
$$
= \tilde{\mu}(x) + \tilde{K}(x, x^{(1:n)} \cup z^{(1:q)}) \left( \tilde{K}(x^{(1:n)} \cup z^{(1:q)}, x^{(1:n)} \cup z^{(1:q)}) \right.
$$
$$
\left. + \operatorname{diag}\{\sigma^2(x^{(1:n)} \cup z^{(1:q)})\} \right)^{-1} \left( (y, \nabla y)(x^{(1:n)} \cup z^{(1:q)}) - \tilde{\mu}(x^{(1:n)} \cup z^{(1:q)}) \right).
$$

Then condition (2) holds only if $\tilde{K}\left( x^*(y(z^{(1:q)}), \nabla y(z^{(1:q)})), x^{(1:n)} \cup z^{(1:q)} \right)$ is constant for all $\nabla y(z^{(1:q)})$ (in a conditionally almost sure set $\mathcal{S}_2$ allowed to depend on $y(z^{(1:q)})$), which holds only if $x^*(y(z^{(1:q)}), \nabla y(z^{(1:q)}))$ are the same for all $\nabla y(z^{(1:q)}) \in \mathcal{S}_2$. Thus, the inequality is strict in settings where $\nabla y(z^{(1:q)})$ affects $x^*(y(z^{(1:q)}), \nabla y(z^{(1:q)}))$. $\qquad\square$

Next we analyze the Bayesian optimization problem under a dynamic programming (DP) framework and show that *d-KG* is one-step Bayes-optimal.

*Proof of Proposition 2.* Suppose that we are given a budget of $N$ samples, i.e. we may run the algorithm for $N$ iterations. Our goal is to choose sampling decisions ($\{z^i, 1 \leq i \leq Nq\}$ and the implementation decision $z^{Nq+1}$ that minimizes $f(z^{Nq+1})$. We assume that $(f(x), \nabla f(x))$ is drawn from the prior $\mathcal{GP}(\tilde{\mu}, \tilde{K})$, then $(f(x), \nabla f(x))$ follows the posterior process $\mathcal{GP}(\tilde{\mu}^{(Nq)}, \tilde{K}^{(Nq)})$ after $N$ iterations, so we have $\mathbb{E}_{Nq}(f(z^{Nq+1})) = \tilde{\mu}_1^{(Nq)}(z^{Nq+1})$. Thus, letting $\Pi$ be the set of feasible policies $\pi$, we can formulate our problem as follows

$$
\inf_{\pi \in \Pi} \mathbb{E}^\pi \left[ \min_{x \in \mathbb{A}} \tilde{\mu}_1^{(Nq)}(x) \right].
$$

We analyze this problem under the DP framework. We define our state space as $S^n := (\tilde{\mu}^{(nq)}, \tilde{K}^{(nq)})$ after iteration $n$ as it completely characterizes our belief on $f$. Under the DP framework, we will define the value function $V^n$ as follows

$$
V^n(s) := \inf_{\pi \in \Pi} \mathbb{E}^\pi \left[ \min_{x \in \mathbb{A}} \tilde{\mu}_1^{(Nq)}(x) \big| S^n = s \right] \tag{B.2}
$$

for every $s = (\mu, K)$. The Bellman equation tells us that the value function can be written recursively as

$$
V^n(s) = \min_{\boldsymbol{z} \in \mathbb{A}^q} Q^n(s, \boldsymbol{z})
$$

where

$$
Q^n(s, \boldsymbol{z}) = \mathbb{E}\left[ V^{n+1}(S^{n+1}) | S^n = s, z^{((nq+1):(n+1)q)} = \boldsymbol{z} \right]
$$

At the same time, we also know that any policy $\pi^*$ whose decisions satisfy

$$
Z^{\pi^*, n}(s) \in \operatorname{argmin}_{\boldsymbol{z} \in \mathbb{A}^q} Q^n(s, \boldsymbol{z}) \tag{B.3}
$$

is optimal. If we were to stop at iteration $n+1$, then $V^{n+1}(S^{n+1}) = \min_{x \in \mathbb{A}} \tilde{\mu}_1^{((n+1)q)}(x)$ and (B.3) reduces to

$$
Z^{\pi^*, n}(s) \in \operatorname{argmin}_{\boldsymbol{z} \in \mathbb{A}^q} \mathbb{E}\left[ \min_{x \in \mathbb{A}} \tilde{\mu}_1^{((n+1)q)}(x) \mid S^n = s, z^{((nq+1):(n+1)q)} = \boldsymbol{z} \right]
$$
$$
= \operatorname{argmax}_{\boldsymbol{z} \in \mathbb{A}^q} \left\{ \min_{x \in \mathbb{A}} \tilde{\mu}_1^{(nq)}(x) - \mathbb{E}\left[ \min_{x \in \mathbb{A}} \tilde{\mu}_1^{((n+1)q)}(x) \mid S^n = s, z^{((nq+1):(n+1)q)} = \boldsymbol{z} \right] \right\},
$$

which is exactly the *d-KG* algorithm. This proves that *d-KG* is one-step Bayes-optimal. $\qquad\square$

# C   Proof of Theorem 1

Recall that we define the value function in Eq. (B.2). Similarly, we can define the value function for a specific policy $\pi$ as

$$V^{\pi,n}(s) := \mathbb{E}^\pi \left[ \min_{x \in \mathbb{A}} \tilde{\mu}_1^{(Nq)}(x) | S^n = s \right]. \tag{C.1}$$

Since we are varying the number of iterations $N$, we define $V^0(s; N)$ as the optimal value function when the size of the iteration budget is $N$. Additionally, we define $V(s; \infty) := \lim_{N \to \infty} V^0(s; N)$. Similarly, we define $V^{0,\pi}(s; N)$ and $V^\pi(s; \infty)$ for a specific policy $\pi$.

Next we will state two lemmas concerning the benefits of additional samples, which will be useful in the latter proofs. First we have the following result for any stationary policy $\pi$. A policy is called *stationary* if the decision of the policy only depends on the current state $S^n := (\tilde{\mu}^{(n)}, \tilde{K}^{(n)})$ (not on the iteration $n$). *d-KG* is stationary.

**Lemma 1.** *For any stationary policy $\pi$ and state $s$, $V^{\pi,n}(s) \leq V^{\pi,n+1}(s)$.*

This lemma states that for any stationary policy, one additional iteration helps on average.

*Proof of Lemma 1.* We prove by induction on $n$. When $n = N - 1$, by Jensen's inequality,

$$
\begin{aligned}
V^{\pi,N-1}(s) &= \mathbb{E}^\pi \left[ \min_x \tilde{\mu}_1^{(Nq)}(x) \mid S_{N-1} = s \right] \\
&\leq \min_x \mathbb{E}^\pi \left[ \tilde{\mu}_1^{(Nq)}(x) \mid S_{N-1} = s \right] \\
&= V^{\pi,N}(s).
\end{aligned}
$$

Then by the induction hypothesis,

$$
\begin{aligned}
V^{\pi,n}(s) &= \mathbb{E}^\pi \left[ V^{\pi,n+1}(S^{n+1}) | S^n = s \right] \\
&\leq \mathbb{E}^\pi \left[ V^{\pi,n+2}(S^{n+1}) | S^n = s \right] \\
&= V^{\pi,n+1}(s),
\end{aligned}
$$

where line 2 above is due to the induction hypothesis and line 3 is due to the stationarity of the policy and the transition kernel of $\{S^n : n \geq 0\}$. We conclude the proof. $\square$

The following lemma is related to the optimal policy. It says that if allowed an extra fixed batch of samples, the optimal policy performs better on average than if no extra samples allowed.

**Lemma 2.** *For any state $s$ and $z \in \mathbb{A}$, $Q^n(s, x) \leq V^{n+1}(s)$.*

As a direct corollary, we have $V^n(s) \leq V^{n+1}(s)$ for any state $s$.

*Proof of Lemma 2.* The proof of Lemma 2 is quite similar to that of Lemma 1. We omit the details here. $\square$

Recall that $V(s; \infty) := \lim_{N \to \infty} V^0(s; N)$. The lemma below shows that $V(s; \infty)$ is well defined and bounded below.

**Lemma 3.** *For any state $s$, $V(s; \infty)$ exists and*

$$V(s; \infty) \geq U(s) := \mathbb{E} \left[ \min_{x \in \mathbb{A}} f(x) | S^0 = s \right]. \tag{C.2}$$

*Proof of Lemma 3.* We will show that $V^0(s; N)$ is non-increasing in $N$ and bounded below from $U(s)$. This will imply that $V^0(s; \infty)$ exists and is bounded below from $U(s)$. To prove that $V^0(s; N)$ is non-increasing of $N$, we note that

$$
\begin{aligned}
&V^0(s; N) - V^0(s; N - 1) \\
&= V^0(s; N) - V^1(s; N) \\
&\leq 0,
\end{aligned}
$$

by the corollary of Lemma 2. To show that $V^0(s; N)$ is bounded below by $U(s)$, for every $N \geq 1$ and policy $\pi$,

$$
\begin{aligned}
\mathbb{E}^\pi \left[ \min_x \tilde{\mu}_1^{(Nq)}(x) | S^0 = s \right] &= \mathbb{E}^\pi \left[ \min_x \mathbb{E}_N^\pi \left[ f(x) \right] | S^0 = s \right] \\
&\geq \mathbb{E}^\pi \left[ \mathbb{E}_N^\pi \left[ \min_x f(x) \right] | S^0 = s \right] \\
&= \mathbb{E}^\pi \left[ \min_x f(x) | S^0 = s \right] \\
&= \mathbb{E} \left[ \min_x f(x) | S^0 = s \right] = U(s).
\end{aligned}
$$

Thus we have $V^0(s; N) \geq U(s)$. Taking the limit $N \to \infty$, we have $V(s, \infty) \geq U(s)$.

Similarly, we can show that $V^{\pi,0}(s; N)$ is non-increasing in $N$ and bounded below from $U(s)$ for any stationary policy $\pi$ by exploiting Lemma 1. This implies that $V^\pi(S^0; \infty)$ exists for each stationary policy. □

For a policy $\pi$, $V^\pi(s, \infty) = U(s)$ means that the policy $\pi$ can successfully find the minimum of the function if the function is sampled from a GP with the mean and the kernel given by $s$. The following lemma is the key to prove asymptotic consistency. Recall that in Theorem 1, we assume that $\mathbb{A}$ is finite. When $\mathbb{A}$ is finite, we have

**Lemma 4.** *If a stationary policy $\pi$ measures every alternative $x \in \mathbb{A}$ infinitely often almost surely in the noisy case or $\pi$ measures every alternative $x \in \mathbb{A}$ at least once in the noise-free case, then $\pi$ is asymptotically consistent and has value $U(s)$.*

*Proof of Lemma 4.* We assume that the measurement noise is of finite variance, it implies that the posterior sequence $\tilde{\mu}_1^{(Nq)}$ converges to the true surface $f$ by the vector-version strong law of large numbers if we sample every alternative infinitely often in the noisy case or at least once in the noise-free case. Thus, $\lim_{N \to \infty} \mu^{(Nq)} = f$ a.s., and $\lim_{N \to \infty} \min_{x \in \mathbb{A}} \tilde{\mu}_1^{(Nq)}(x) = \min_{x \in \mathbb{A}} f(x)$ in probability. Next we will show that $\min_{x \in \mathbb{A}} \tilde{\mu}_1^{(Nq)}(x)$ is uniformly integrable in $N$, which implies that $\min_{x \in \mathbb{A}} \tilde{\mu}_1^{(Nq)}(x)$ converges in $L_1$. For any fixed $K \geq 0$, we have

$$
\begin{aligned}
&\mathbb{E} \left[ \left| \min_{x \in \mathbb{A}} \tilde{\mu}_1^{(Nq)}(x) \right| 1_{\{| \min_{x \in \mathbb{A}} \tilde{\mu}_1^{(Nq)}(x)| \geq K\}} \right] \\
\leq\ & \mathbb{E} \left[ \max_{x \in \mathbb{A}} \left| \tilde{\mu}_1^{(Nq)}(x) \right| 1_{\{\max_{x \in \mathbb{A}} |\tilde{\mu}_1^{(Nq)}(x)| \geq K\}} \right] \\
=\ & \mathbb{E} \left[ \max_{x \in \mathbb{A}} |\mathbb{E}_{Nq}(f(x))| 1_{\{\max_{x \in \mathbb{A}} |\mathbb{E}_{Nq}(f(x))| \geq K\}} \right] \\
\leq\ & \mathbb{E} \left[ \max_{x \in \mathbb{A}} \mathbb{E}_{Nq}(|f(x)|) 1_{\{\max_{x \in \mathbb{A}} \mathbb{E}_{Nq}(|f(x)|) \geq K\}} \right] \\
\leq\ & \mathbb{E} \left[ \mathbb{E}_{Nq}(\max_{x \in \mathbb{A}} |f(x)|) 1_{\{\mathbb{E}_{Nq}(\max_{x \in \mathbb{A}} |f(x)|) \geq K\}} \right] \\
=\ & \mathbb{E} \left[ \mathbb{E}_{Nq} \left( \max_{x \in \mathbb{A}} |f(x)| 1_{\{\mathbb{E}_{Nq}(\max_{x \in \mathbb{A}} |f(x)|) \geq K\}} \right) \right] \\
=\ & \mathbb{E} \left[ \max_{x \in \mathbb{A}} |f(x)| 1_{\{\mathbb{E}(\max_{x \in \mathbb{A}} |f(x)|) \geq K\}} \right].
\end{aligned}
$$

Since $\max_{x \in \mathbb{A}} |f(x)|$ is integrable and $P(\max_{x \in \mathbb{A}} |f(x)|) \geq K) \leq \mathbb{E}(\max_x |f(x)|)/K$ is bounded uniformly in $N$ and goes to zero as $K$ increases to infinity, Given that $\min_{x \in \mathbb{A}} \tilde{\mu}_1^{(Nq)}(x)$ converges

in $L_1$, we have

$$
\begin{aligned}
V^\pi(s;\infty) &= \lim_{N\to\infty} \mathbb{E}^\pi[\min_{x\in\mathbb{A}} \tilde{\mu}_1^{(Nq)}(x)|S^0 = s] \\
&= \mathbb{E}^\pi[\lim_{N\to\infty} \min_{x\in\mathbb{A}} \tilde{\mu}_1^{(Nq)}(x)|S^0 = s] \\
&= \mathbb{E}^\pi[\min_{x\in\mathbb{A}} f(x)|S^0 = s] \\
&= U(s).
\end{aligned}
$$

By Lemma 3 above, we conclude that $V^\pi(s;\infty) = V(s;\infty) = U(s)$. $\qquad\square$

Then we will show that *d-KG* measures every alternative $x \in \mathbb{A}$ infinitely often in the noisy case or *d-KG* measures every alternative $x \in \mathbb{A}$ at least once when $N$ goes to infinity, which leads to the proof of Theorem 1.

*Proof of Theorem 1.* We focus on the noisy case for clarity. One can provide an identical proof for the noise-free case by replacing sampling infinitely often with sampling at least once.

By a similar proof with Lemma A.5 in Frazier et al. [9], we can show that $S^n$ converges to a random variable $S^\infty := (\tilde{\mu}^\infty, \tilde{K}^\infty)$ as $n$ increases. By definition,

$$
\begin{aligned}
&V^N(S^\infty) - Q^{N-1}(S^\infty; z^{(1:q)}) \\
={}& \min_x \tilde{\mu}_1^\infty(x) - \mathbb{E}\left[\min_x \left(\tilde{\mu}_1^\infty(x) + e_1^T \tilde{\sigma}^\infty(x, z^{(1:q)}) Z_{q(d+1)}\right)\right] \\
\geq{}& \min_x \tilde{\mu}_1^\infty(x) - \mathbb{E}\left[\min_{x\in z^{(1:q)}} \left(\tilde{\mu}_1^\infty(x) + e_1^T \tilde{\sigma}^\infty(x, z^{(1:q)}) Z_{q(d+1)}\right)\right]
\end{aligned}
$$

If we have measured $z^{(1)}, \cdots, z^{(q)}$ infinitely often in the noisy case, there will be no uncertainty around $f(z^{(1:q)})$ in $S^\infty$, then $V^N(S^\infty) = Q^{N-1}(S^\infty; z^{(1:q)})$. Otherwise $V^N(S^\infty) > Q^{N-1}(S^\infty; z^{(1:q)})$, i.e. there are benefits measuring $z^{(1:q)}$. We define $E = \{x \in \mathbb{A} :$ the number of times measuring $x < \infty\}$, then for any $x \in E$ and $y^{(1:q)} \subset E^c$, we have $Q^{N-1}(S^\infty; z^{(1:(q-1))} \cup x) < V^N(S^\infty) = Q^{N-1}(S^\infty; y^{(1:q)})$. By the definition of *d-KG*, it will measure some $x \in E$, i.e. at least one of $x$ in $E$ is measured infinitely often, a contradiction. $\qquad\square$