[Reviews · NeurIPS 2017]

Reviewer 1



Summary of the paper: This paper describes a technique for incorporating gradient information into Bayesian optimization. The proposed method not only considers gradient information in the modelling process. That is, in the GP that is used to model the objective. But also in the computation of the acquisition function. This is done by using the knowledge gradient method, which measures the expected improvement in the posterior mean of the GP at the observed locations. The proposed approach is shown to be Bayes-optimal and asymptotically consistent. Several experiments including synthetic functions, and real-world experiments involving kernel learning and deep learning illustrate the benefits of the proposed method. Detailed comments: Quality: I think the quality of the paper is high. It is very well written, with the exception of a few typos, and it is supported by both theoretical and empirical results. The paper also reviews important related methods and has an exhaustive experimental section comparing with them. Clarity: The paper is clearly written and important information is included in the supplementary material, including the proofs of the propositions and theorems considered. Something strange is the non-smoothness of the EI function in Figure 1. Why is this so? Is it related to the noisy assumption? Originality: As indicated by the authors, considering gradient information is not new. Some authors have already considered this. However, and importantly, it seems that the technique proposed by the authors is the only one that considers this information in the computation of the acquisition function. Previous methods seem to consider this information only in the modeling of the objective. It is also a very interesting idea to use stochastic gradient optimization to maximize the acquisition function. As far as I know this is the first work considering that. Significance: I think this is an interesting paper for the community working on Bayesian optimization. It will certainly receive some attention. The experimental results also suggest important gains by considering gradient information, and they look significant. Something that can be criticized is that obtaining gradient information will probably be more expensive. Thus, gradient free methods will be able to do more evaluations of the objective given a similar computational time. This has not been considered by the authors.

Reviewer 2



Major: To me this paper is an important one as it presents substantial contributions to Bayesian Optimization that are at the same time elegant and practically efficient. While the paper is well written, it could be formally improved in places; below are a few comments questions that might be useful toward such improvements. Besides this, I was a bit puzzled by the fact that Theorem 1 is not very precisely stated; also, given the fact that it is proven in appendix (which is understandable given the space limitations) and with a proof that requires substantial time to proofread formally (as it is almost a second paper in itself, with a specific policy formalism/set-up), I had to adapt my confidence score accordingly. Also, I was wondering if the consistency proof could work in the case of noiseless observations, where it does not make sense to replicate evaluations at the same point(s)? Nevertheless, from the methodological contributions and their practical relevance, I am conviced that the presented approach should be spread to BO/ML researchers and that NIPS is an excellent platform to do so. Minor: * In the abstract: "most optimization methods" is a bit strong. Derivative-free optimization methods should not be forgotten. Yet, gradients are indeed essential in (differentiable) local optimization methods. * In the abstract: syntax issue in "We show d-KG provides". * Introduction: from "By contrast", see comment above regarding *local* optimization. * Introduction: "relative value" does not sound very clear to me. Maybe speak of "variations" or so? * Introduction: "we explore the what, when, and why". In a way yes, but the main emphasis is on d-KG, and even if generalities and other approaches are tackled, it is probably a bit strong to claim/imply that *Bayesian Optimization with Gradients* is *explored* in a paper subjected to such space restrictions. * Introduction: from "And even" to "finite differences"; in such case, why not use this credit to do more criterion-driven evaluations? Are there good reasons to believe that two neigbouring points approximating a derivative will be better than two (more) distant well-chosen points? [NB: could there exist a connection to "twin points", in Gaussian kernel settings?] * Section 3.1: "to find argmin"; so, the set [in case of several minimizers]? * Section 3.1: "linear operator" calls for additional regularity condition(s). By the way, it would be nice to ensure differentiability of the GP via the chosen mu and K. * Section 3.1: the notation used in equation (3.2) seem to imply that there cannot be multiple evaluations at the same x. Notations in terms of y_i (and similar for the gradient), corresponding to points x_i, could accomodate that. * Section 3.1: the paragraph from line 121 to 126 sounds a bit confusing to me (I found the writing rather cryptic there) * Section 3.2: "expected loss"; one would expect a translation by the actual min there (so that loss=0 for successful optim) * Section 3.2: in equation (3.3), conditioning on "z" is abusive. By the way, why this "z" notation (and z^(i)) for "x" points? * Algorithm 2, lines 2: a) =argmax abusive and b) is theta on the unit sphere? restricted to canonical directions? * Section 3.3: in the expression of the d-KG factor, the term \hat{\sigma}^{(n)} appears to be related to (a generalization of) kriging update formulas for batch-sequential data assimilation. Maybe it could make sense to connect one result ro the other? * Section 3.3: the approach with the enveloppe theorem is really nice. However, how are the x^*(W) estimated? Detail needed... * Section 3.3: "K samples"; from which distribution? + Be careful, the letter K was already used before, for the covariance. * Section 3.3, line 191: "conditioning" does not seem right here...conditional? conditionally? * Section 3.4: in Prop.1, you mean for all z^{(1:q)}? Globally the propositions lack mathematical precision... * Section 3.4: ...in particular, what are the assumptions in Theorem 1? E.g., Is f assumed to be drawm from the running GP? * Section 4: notation Sigma is used for the covariance kernel while K was used before. * Section 4: about "the number of function evaluations"; do the figures account for the fact that "d-approaches" do several f evaluations at each iteration? * Section 5: about "low dimensional"; not necessarily...See REMBO and also recent attempts to combine BO and kernel (sparse) decompositions.

Reviewer 3



The paper proposes a novel acquisition function d-KG (for derivative-enabled knowledge-gradient) which is an extension of the knowledge-gradient (KG) by Wu and Frazier 2016. The main difference, to my understanding, is that the possible future gradients (batch as well as online evaluations) also affect the acquisition function, and not just the marginal posterior over the the objective, which the authors argue, should improve the selection of evaluation-locations. The authors also provide a way of estimating the gradients of d-KG in an unbiased way, such that stochastic optimizers can be used to maximize it. The method is evaluated on some classic synthetic benchmarks of BO and real work applications e.g. a logistic regressors and an MLP. The paper is well written and clear. The novel contributions are clearly marked as such and the connections and distinctions to existing methods that authors point out are very informative. I did not entirely follow the proof of Section 3.4, nor the derivation of \nabla d-KG since they were not contained in the main paper, but the line of argument contained in 3.3 sounds plausible to me. Some questions: - why is the noise on the derivative observations independent? likewise the independence between gradient and value observation. Is this purely for computational convenience/easier estimation of \sigma? or is there an underlying reason? - I am not quite sure how \theta is defined. Is \theta \nabla y (below Eq. 3.5) a projection? If yes, a transpose is missing. Also, must \theta have unit norm? - In Section 4.1, why did you observed the fourth derivative? From pseudocode 1 it looks as \theta is optimized for. A major practical concern: - You use SGD to optimize d-KG. How do you tune its learning rate? This might be quite tricky in practice, especially since you might need to re-tune it after each update of the GP and not just one time for the whole BO-procedure. Nitpicky comments: - line 168: is it acontuoussteadofan continuous'? - y-labels on Figure 2 would be nice. - line 242: Weals..:wordmissing?-le288:are can': one word too much? - please don't call a shallow MLP with 2 hidden layers `deep neural network' (lines 283 and 292 and the conclusion) (*Edit:* I increased the score since I found the arguments of the authors convincing. But I encourage the authors to comment on the learning rate tuning as they promised in the rebuttal.)